# Vaginal Reconstruction in Patients with Mayer–Rokitansky–Küster–Hauser Syndrome—One Centre Experience

**DOI:** 10.3390/medicina56070327

**Published:** 2020-07-01

**Authors:** Adelaida Avino, Laura Răducu, Adrian Tulin, Daniela-Elena Gheoca-Mutu, Andra-Elena Balcangiu-Stroescu, Cristina-Nicoleta Marina, Cristian-Radu Jecan

**Affiliations:** 1Department of Plastic and Reconstructive Surgery, “Prof. Dr. Agrippa Ionescu” Clinical Emergency Hospital, 011356 Bucharest, Romania; adelaida.avino@gmail.com (A.A.); mutu.danielaa@gmail.com (D.-E.G.-M.); cristina.cozma88@yahoo.com (C.-N.M.); jecan.radu@gmail.com (C.-R.J.); 2Doctoral School, “Carol Davila” University of Medicine and Pharmacy, Faculty of Medicine, 020021 Bucharest, Romania; 3Department of Plastic and Reconstructive Surgery, Faculty of Medicine “Carol Davila” University of Medicine and Pharmacy, 020021 Bucharest, Romania; 4Department of General Surgery, “Prof. Dr. Agrippa Ionescu” Clinical Emergency Hospital, 011356 Bucharest, Romania; dr_2lin@yahoo.com; 5Department of Anatomy, Faculty of Medicine, “Carol Davila” University of Medicine and Pharmacy, 020021 Bucharest, Romania; 6Department of Dialysis, Emergency University Hospital, 050098 Bucharest, Romania; stroescu_andra@yahoo.ro; 7Discipline of Physiology, Faculty of Dental Medicine,“Carol Davila” University of Medicine and Pharmacy, 020021 Bucharest, Romania

**Keywords:** Mayer–Rokitansky–Küster–Hauser syndrome, primary amenorrhea, surgical management, vaginal reconstruction, plastic surgery

## Abstract

*Background and Objectives:* The Mayer–Rokitansky–Küster–Hauser syndrome is a congenital condition in which patients are born with vaginal and uterus agenesis, affecting the ability to have a normal sexual life and to bear children. Vaginal reconstruction is a challenging procedure for plastic surgeons. The aim of this study is to report our experience in the management of twelve patients with congenital absence of the vagina due to the MRKH syndrome. *Materials and Methods:* We performed a retrospective study on 12 patients admitted to the Plastic Surgery Department of the Clinical Emergency Hospital “Prof. Dr. Agrippa Ionescu”, Bucharest, Romania, for vaginal reconstruction within a period of eleven years (January 2009–December 2019). All patients were diagnosed by the gynaecologists with vaginal agenesis, as part of the Mayer–Rokitansky–Küster–Hauser syndrome. The Abbe‘–McIndoe technique with an autologous skin graft was performed in all cases. *Results:* The average age of our patients was 20.16 (16–28) years. All patients were 46 XX. The average surgical timing was 3.05 h (range 2.85–4h). Postoperative rectovaginal fistula was encountered in 1 patient. Postoperative average vaginal length was 10.4 cm (range 9.8–12.1 cm). Regular sexual life was achieved in 10 patients. *Conclusion:* Nowadays, there is no established standard method of vaginal reconstruction. In Romania, the McIndoe technique is the most applied. Unfortunately, even if the MRKH syndrome is not uncommon, less and less surgeons are willing to perform the procedure to create a neovagina.

## 1. Introduction

The Mayer–Rokitansky–Küster–Hauser (MRKH) syndrome is considered to be the second most frequent cause of primary amenorrhea [1] being characterized by congenital absence of the uterus, and the upper part (2/3 proximal) of the vagina. The patients have a normal 46XX karyotype with a physiological growth of the secondary sexual features [2]. In the general population, the prevalence is up to 0.02% [3]. This syndrome is not linked with any racial predisposition. Even if it is a congenital disease, in most cases MRKH could not be diagnosed up to adolescence or at the beginning of adulthood [4].

There are two types described in the literature. The first one is characterised by solitary absence of the proximal two-thirds of the vagina [5] and agenesis of the uterus [4], although two symmetric rudimentary horns are present. They are linked by a peritoneal fold to the fallopian tubes, whereas the ovaries and the renal system have a normal development. This appears in almost 44% of all cases. There has been a reported growth of abnormalities of the caudal part of the Müllerian ducts. No other congenital malformations is detected, the patients are utterly asymptomatic and diagnosed during late adolescence due to primary amenorrhea [1]. Meanwhile, type II is associated with other congenital defects including vertebral, cardiac, auditory, renal and vertebral malformations [5]. This affects up to 56% of the patients, involving asymmetrical hypoplasia of one or two buds, with or without dysplasia of the fallopian tubes [1].

Precise history, clinical evaluation and transabdominal ultrasonography are used to diagnose the MRKH syndrome. After this, a check-up of the renal, skeletal, auditory and cardiac systems must be performed. Ovarian function can be analysed through the serum levels of the follicle stimulating hormone, luteinizing hormone, 17ß-oestradiol and androgens [6].

The optimal treatment is vaginal reconstruction in physically and psychologically mature women who are ready to start sexual intercourse [7]. In the literature, a variety of techniques have been presented, each with its advantages and disadvantages. Most commonly, the neovagina is created within the rectovesical space lined with skin (McIndoe Reed technique), peritoneum (Davydov procedure) or intestine. It is possible that in the future, vaginal mucosa-like engineered tissue will be used [8]. Moreover, an important step of therapeutic management is the psychosocial assistance not only for the patients, but also for their parents [9].

The aim of this study is to report our experience in the management of twelve patients with congenital absence of the vagina due to the MRKH syndrome.

## 2. Materials and Methods

We conducted a retrospective study on 12 patients admitted to the Plastic Surgery Department of the Clinical Emergency Hospital “Prof. Dr. Agrippa Ionescu”, Bucharest, Romania, for vaginal reconstruction within a period of eleven years (January 2009–December 2019). Local ethical agreement and informed consent of the patients were obtained. The number of the document from the Ethical Commission of Clinical Emergency Hospital “Prof. Dr. Agrippa Ionescu” is 104663/10.04.2020. All patients were diagnosed by gynaecologists with vaginal agenesis, as part of the Mayer–Rokitansky–Küster–Hauser syndrome. All the data were taken from surgical operating files, medical letters and postoperative records. The data comprised demographic information, chromosomal analysis, surgery timing, preoperative and postoperative vaginal length, complications, postoperative treatment, but also other congenital malformations. The marital status was also evaluated. All cases underwent clinical examination and pelvic ultrasonography during preoperative evaluation. The surgical team included surgeons from the department of general surgery and plastic and reconstructive surgery. Preoperatively, a combination of amoxicillin and clavulanic acid was administered as a prophylactic antibiotic.

The Abbe‘–McIndoe technique with an autologous skin graft was performed in all cases. The surgical procedure was performed on patients placed in the lithotomy position (Figure 1) and under general anaesthesia with urinary catheterization. The general surgeon initiated the intervention. Through a Y-shaped incision in the perineum, a vesico-rectal cavity was created (Figure 2). The dissection was performed between the urethra, bladder and the rectum, the Douglas pouch being the upper and posterior limit. A careful haemostasis of the neovaginal cavity was the last manoeuvre performed by the general surgeon. A split-thickness skin graft was harvested from the anterior part of the thigh, using a Humby knife. The graft was placed on a vaginal stent with its inner surface, the dermal part facing towards the exterior and was sewn with non-absorbable sutures (Figure 3).

Afterwards, the stent was inserted into the preformed neovaginal space. The free margins of the skin graft were sutured to the borders of the incisions made in the perineum, creating this way the introitus of the neovagina. In all cases, the vaginal stent was made of silicone, having the shape of a cylinder with a channel inside it, in order to cleanse the neovagina. The patients were put on absolute bed rest and a special diet was given for the first 9 days. The neovagina was irrigated daily through the channel of the stent with diluted (1%) povidone-iodine solution and normal saline. After 9 days, the vaginal stent was removed and the neovagina was checked and cleansed. Elastic panties were worn in order to retain the stent. The sutures were eliminated after 14 days (Figure 4). In the first 3 months, the vaginal stent had to be kept permanently, except when using the toilet or while bathing, in order to prevent the contraction of the vagina. During this period, the patients irrigated the vagina with sodium bicarbonate solution, wormwood tea or a mixture of normal saline and povidone-iodine. Patients were informed not to replace the vaginal stent into the vagina unless it had been washed with soap and hot water and they had applied cream with the free protein extract of calf blood or vitamin A oil. Subsequently, they were advised using the vaginal stent only during the night for the next 3 months. Six months after the surgery, the vaginal stent had to be used for 3 nights per week, until the patient had a stable partner and regular sexual intercourse (at least 3 times per week). In the case of the absence of regular sexual intercourse, the stent had to be used 3 nights per week to maintain the vaginal cavity open.

Postoperatively, all patients were followed up monthly in the first 6 months, respectively, 9 months and after that, once a year.

Local ethical agreement and informed consent of the patients were obtained.

## 3. Results

Twelve cases of vaginal reconstruction were performed throughout the period of the study. The average age of the patients undergoing surgery was 20.16 years (range 16–28 years). All the patients were living in an urban area and two of them were smokers. Preoperatively, one patient was engaged, and the others were single, but after the surgery ten patients found a stable partner and got married.

All patients presented primary amenorrhea at 14–15 years old. Upon clinical examination, external genitalia had a normal appearance in all patients. None of them presented a uterus, but only ovaries. The vagina was totally absent in all cases. According to chromosomal analysis, all patients were 46XX. One patient had an anorectal malformation and two interventions were performed by a paediatric surgeon during childhood. Moreover, she also presented mitral valve regurgitation. One patient, a 25-year-oldwoman came to our clinic with a neovagina of 3 cm, with the complaint that she could not have a normal sexual life with her future husband. Her first vaginal reconstruction was performed when she was 22. The modified McIndoe technique with full-thickness skin grafts harvested from the anterior part of the thigh was performed in all cases. The average surgical timing was 3.05 h (range 2.85–4 h). Postoperatively, the vaginal length varied from 9.8 to 12.1 cm, with a mean length of 10.4 cm.

Regarding acute postoperative complications, we would like to mention the acute bleeding of the donor site in one patient. Moreover, haemorrhage of the posterior wall was encountered in one case, 12 days after the surgery. The patient raised the skin graft from the posterior wall as a result of forcing the stent inside the neovagina after it slipped out during the night. Meticulous haemostasis was decided and re-epithelialization of the affected part was preferred from the integrated skin grafted margins. Silver dressings were used to cover the donor site for 10 days.

The surgical intervention performed for the patient with the neovagina of 3 cm was challenging. The space that was previously created became a fibrotic scar tissue, which was difficult to dissect. After one month postoperative, the patient presented rectovaginal fistula. Colostomy was decided before repairing the fistula. The colostomy was kept for 6 months, until the fistula was completely healed.

No early complications were seen in other cases, and all of them used a vaginal stent, as recommended. During the follow-up, 4 of them presented keloid scars on the donor site and recurrent urinary tract infections were detected in 9 patients. Psychosocial assistance was mandatory for all the patients.

## 4. Discussion

The Mayer–Rokitansky–Küster–Hauser syndrome is the most common cause of vaginal agenesis, known also as Müllerian agenesis, Müllerian aplasia or CAUV (congenital absence of the uterus and vagina) [10]. It is not considered a rare disease, but it is usually discovered in adolescence due to primary amenorrhea [11]. More than 2000 years ago, Hippocrates mentioned vaginal agenesis for the first time [12] as “membranous obstruction” (obstructed vagina), in his book “De la nature de la femme” (On the Nature of Women). Not until 1559 were the medical reports of Matteo Realdo Colombo, regarding the absence of both the uterus and vagina, first mentioned in literature. In 1572, he described the disease under the name “vulva rara”. The medical eponym honours August Franz Josef Karl Mayer, Karl Freiherr von Rokitansky, Hermann Küster and Georges Andre Hauser thanks to their remarkable contributions towards the discovery of this congenital syndrome. In 1829, Mayer presented the anatomical abnormalities; in 1838 [10], Rokitansky documented the absence of the uterus and vagina in 19 adult autopsies [13], highlighting the importance of a classification system, based on cases of uterovaginal agenesis. In 1910, Hermann Küster outlined the association of renal and skeletal malformations with the uterovaginal agenesis and in 1961, Hauser presented his findings in 21 patients [10].

Initially, MRKH syndrome was considered to be sporadic, depending on exogenous factors (gestational diabetes) or exposure to teratogens (thalidomide). However, several epidemiological studies failed to identify any relationship between drug use, illness, or exposure to known teratogens during pregnancy and the birth of a child with MRKH [14].Despite many studies that have been conducted to discover the cause, the exact aetiology of the syndrome remains unclear. Defects during embryogenesis lead to malformations of the genitourinary system [4]. Specifically, during the end of the fourth week of fetal life, abnormalities of the intermediate mesoderm determine deficiency in the development of the blastema of the cervicothoracic somites and the pronephric ducts. These modifications affect the mesonephros and then the Wolffian and Müllerian ducts [2].The distal extremity of the Müllerian ducts creates the proximal 2/3rd parts of the vagina and uterine cervix, the intermediate part forms the uterine body, while the cranial segments open in the coelomic cavity (future peritoneal cavity), shaping the fallopian tubes. Meanwhile, the ovaries develop from the mesenchyme and from the epithelium of the genital crest of the intermediate mesoderm, this process is not associated with the mesonephros. Thus, in most cases, the defects of the Müllerian ducts are not linked to abnormalities of the ovaries [4]. The distal third of the vagina is developed from ectodermal cells, so it can create a shallow pouch in the perineum [13]. Regarding genetics, in literature there have been mentions of an association between the mutations of the WNT4 and TCF2 genes and MRKH syndrome [15].

In general, the patients with MRKH syndrome are first seen by a gynaecologist at age 14 to 18 years [16], when the absence of the menarche causes concern. Menstruation does not appear at the usual age because the uterus is absent, but ovulation occurs regularly [13]. However, the patients may be diagnosed at birth or during childhood due to other congenital malformations [16]. In our study, 11 patients were diagnosed at 14–15 years due to primary amenorrhea.

One patient presented a recto-vestibular fistula at birth, which was corrected by the paediatric surgeon. The final diagnosis, the MRKH syndrome, was confirmed during childhood after exploratory laparotomy. It revealed no uterus, but both ovaries were present and attached to a cord-like structure without any evidence of fallopian tubes. Anorectal malformations are exceptionally described as part of the MRKH syndrome, most commonly encountering recto-vestibular fistulae and cloacal malformations [10]. Wang et al. (2010) suggested a single-stage ano-recto-vaginoplasty in patients who had MRKH syndrome and recto-vestibular fistulae with imperforate anuses, to avoid the fibrosis and scarring created by repeated surgery. The surgical intervention should be performed later in life if the recto-vestibular fistula is asymptomatic or causes modest discomfort, and might be guided by dietary adjustments. If the single-stage procedure is performed before the age of 14, long-term postoperative dilation of the neovagina might be difficult for young girls. Therefore, treatment should be decided individually for each patient [17]. In addition, one patient presented a mild mitral valve insufficiency. Heart malformations are less frequent. Pittock et al. (2005) described in their study a mild mitral regurgitation and mitral valve prolapse with valvular regurgitation, but also other cardiac problems in patients with MRKH such as truncus arteriosus with complete repair in infancy and patent ductus arteriosus [18]. The most common heart defects linked to MRKH syndrome are aorto-pulmonary window, atrial septal defects, pulmonary valvular stenosis and tetralogy of Fallot [19].

The primary goal of vaginoplasty in patients with vaginal agenesis is to create a vagina of adequate length, diameter and with a stable lining for sexual intercourse. Several surgical procedures were presented in the literature, from serial dilation, Vecchietti’s technique, sigmoid or ileal flaps, the gracilis flap, the Singapore flap to the expanded vulvar flap [20].

In our study, the McIndoe technique with autologous skin graft was used to create the neovagina. For a better approach, the general surgeon chose a Y-shaped incision in the perineum, instead of H-shaped incision (Bastu et al., 2012) [21]. Due to the fact that our patients presented vaginal agenesis, the surgical intervention was the only option for creating the neovagina. 

In the last 20 years, alternatives of the McIndoe procedure have been proposed because of the permanent scars at the donor site. Lin et al. (2003) highlighted the possibility of covering the vaginal canal with autologous buccal mucosa. Starting from the idea that the autologous buccal mucosa has been used to substitute skin and bladder mucosa grafts in urethroplasty, the procedure was performed on 8 young patients. The results were encouraging, the patients had an adequate vaginal length, without strictures and with no deficiency in opening their mouth. The buccal mucosa was used due to its characteristics of being an easily accessible, non-hair-bearing material, with excellent cosmetic results [22]. Teng et al. (2019) used an autologous micromucosa graft harvested from the vulva and the buccal cavity. The mucosal patch was cut into particles, which were immersed in saline, and then manually adhered to a gauze mould. The 8 cm gauze vaginal stent was changed after 7 days with a 10 cm glass mould, with a satisfactory clinical outcome [23]. Hopefully, in the future, vaginal mucosa-like engineered tissue will be used to line the neovagina. 

Baptista et al. (2016) created the neovagina using the laparoscopic modified Vecchietti technique, an intervention with good results that involves a single suprapubic ancillary trocar. This procedure eliminates the incision of the vesicorectal peritoneum [24]. Unfortunately, in our clinic this technique was never used.

In our study, the vaginal stent was removed after 9 days in all cases. The stent had a 4.5 cm diameter and a length of 13 cm. Seccia et al. (2002) presented their technique, where the removal of the mould was performed after 7 days [25]. Before the daily replacement of the vaginal stent, cream with free protein extract of calf blood was used. This promotes oxidative metabolism and shifts the redox-balance of cells to produce more oxidized substrates, leading to a faster healing [26]. The mean length of the neovagina was 10.4 cm. In the literature, there have been results that presented 7.8 cm [21] or 8.9 cm [20] in length. We did not change or improve the McIndoe procedure, but we used a bigger vaginal stent compared to those reported in the literature, creating in this way a neovagina of 10.4 cm. 

Nowadays, more and more new dressings for wounds are created to reduce pain and local discomfort [27]. We used silver dressings for the donor site to accelerate the healing [28].

In the literature, postoperative complications have been recorded, such as rectovaginal fistula, vaginal stricture, bleeding, recurrent infections of the urinary system, urinary incontinence or rectocele [29,30,31]. In our study, the worst complication that we encountered in one patient was the rectovaginal fistula. All patients presented, years after the interventions, recurrent urinary tract infections.

Even if the patients could adopt a child, none of our patients did. In Romania there is not the possibility of surrogacy or uterine transplantation. 

## 5. Conclusions

Based on our experience, the McIndoe procedure is the classic and optimal intervention procedure for vaginal reconstruction in cases of congenital vaginal agenesis. It is important to highlight the necessity of the proper management of using the vaginal stent postoperatively and of a monthly follow-up in the first 6 months. Moreover, psychosocial assistance is mandatory for all patients.

## Figures and Tables

**Figure 1 medicina-56-00327-f001:**
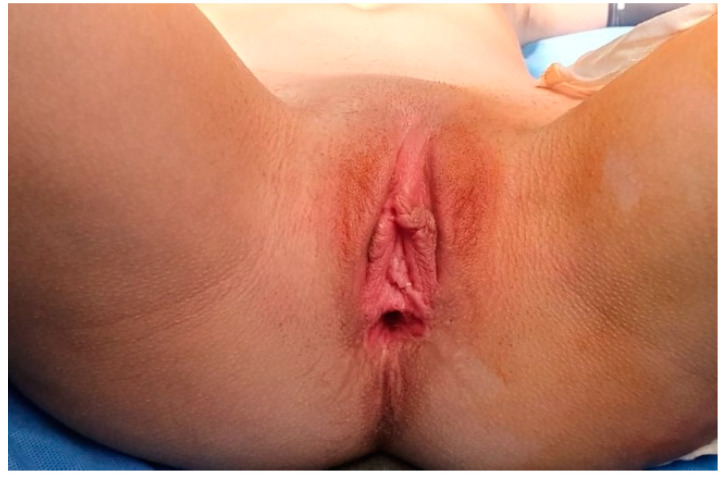
Preoperative photograph in a patient with vaginal agenesis.

**Figure 2 medicina-56-00327-f002:**
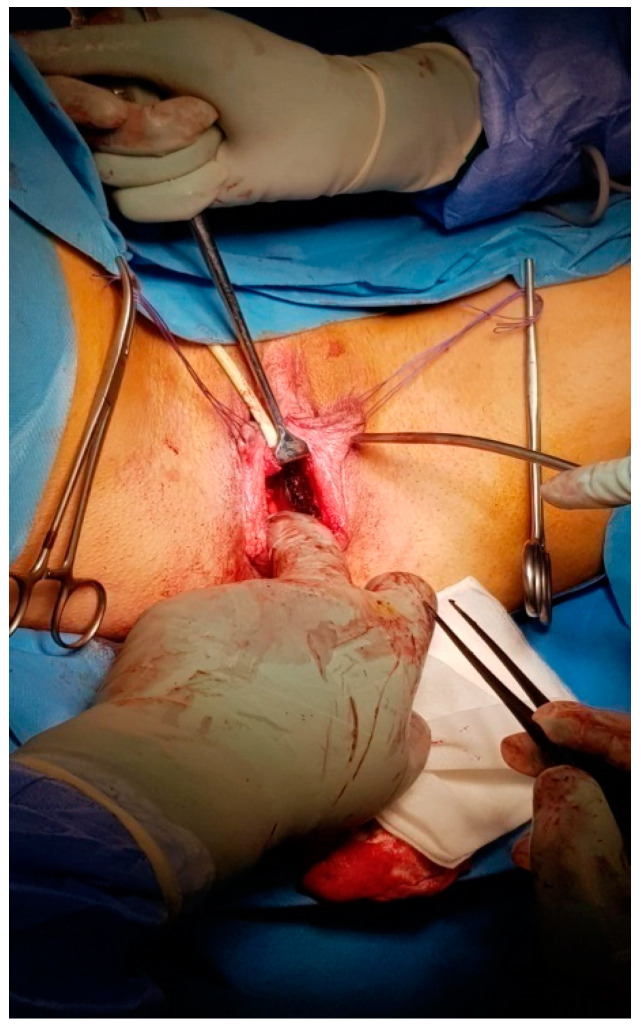
Y-shape incision in the perineum, creating a vesico-rectal cavity.

**Figure 3 medicina-56-00327-f003:**
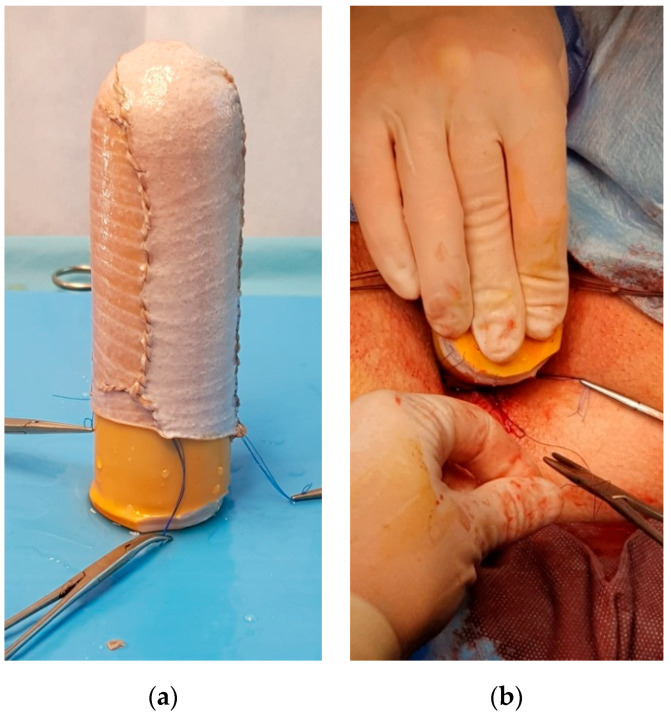
(**a**) The vaginal stent with the skin grafts. (**b**) The fixation of the vaginal stent in the neovaginal space.

**Figure 4 medicina-56-00327-f004:**
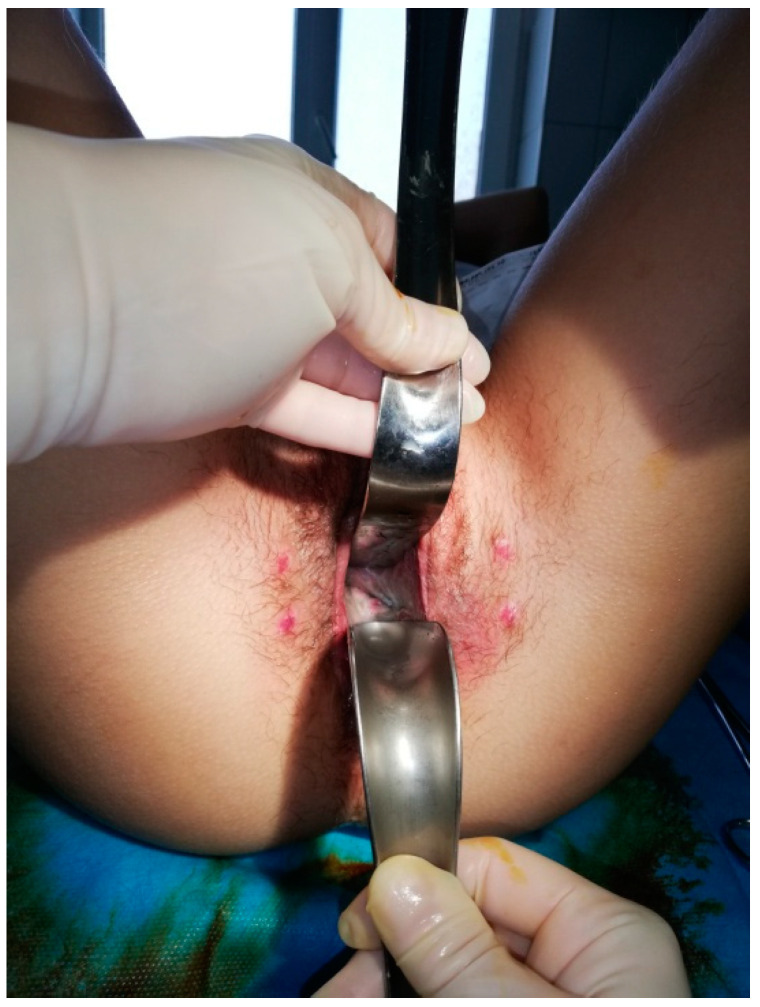
20 days after vaginal reconstruction.

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
