# Peer review of "Vaginal Reconstruction in Patients with Mayer–Rokitansky–Küster–Hauser Syndrome—One Centre Experience"

_medicina, 2020, doi:10.3390/medicina56070327_

Round 1
Reviewer 1 Report
The topics of the article is interesting and relevant. One of the methods of creating a neovagina, the Abbé-McIndoe technique with autologous skin graft, in women with Mayer-Rokitansky-Küster-Hauser syndrome is described.
In Introduction, I would recommend adding more information about other conservative and surgical options for creating a neovagina, especially about the laparoscopic Vecchietti vaginoplasty technique. I would expect that the authors mention reproduction problems of these women and its possible solutions (for example adoption, surrogation and uterine transplantation). In Discussion I would compare advantages and disadvantages of each of conservative and surgical techniques when creating a neovagina and would explain why the authors prefer the Abbé-McIndoe technique.
In the paper, I lack proper description of sexual life and mental problems of the patients before the operation and improvement of these aspects after the neovagina creation, or comparison of their sexuality with control group of healthy women. It is failure of having coitus due to the vaginal agenesis, which is the reason why such operations are performed. When comparing these groups of women, adequate scientific instruments should be used and that are standardized questionnaires and relevant data about their sexuality. Information about surgical aspects of the operation prevail in the article but these are generally known, but I lack assessment of its effect as data about interval of an initiating sexual life, its frequency, sexual satisfaction, reaching orgasm etc.
I recommend revising the article.
Author Response
We truly appreciate the review and consider it useful for other material regarding the vaginal reconstruction but the purpose of this article is not to emphasize the quality of life of these patients, and the sexual satisfaction therefore a questionnaire regarding these aspects was not addressed.
In addition, it is a descriptive article about the technique performed in our clinic and its outcomes. It only contains a short revision of the tecniques described in literature, it is not a comparative study.
I wrote in the manuscript - line 235: Baptista et al. (2016) created the neovagina using laparoscopic modified Vecchietti technique, an intervention with good results that involves a single suprapubic ancillary trocar. This procedure eliminates the incision of the vesicorectal peritoneum. Unfortunately, in our clinic this technique was never used.
Line 253 - Even if the patients can adopt a child, none of our patients did it. In Romania there is not the possibility of surrogation or uterine transplantation.
Reviewer 2 Report
This is an observational case series reporting experience of vaginal reconstruction in 12 patients with MRKH syndrome, performed in a single center, during a period of 10 years. The report is clear and well written describing the technique used, complications and clinical results. The study includes a short review of the literature about MRKH syndrome and the surgical approaches used for creation of neovagina. The authors did not report whether these patients were offered and tried any non surgical approach as vaginal serial dilatation prior to the decision of the surgical approach. Also, as the Mc Indoe procedure is a known surgical technique, with various modifications, it is not clear whether the specific procedure described in this manuscript contains any original addition to the already preivously reported relevant knowledgebase.
Author Response
Thank you bery much for the review.
I wrote in the manuscrip - line 222 -. Due to the fact that our patients presented vaginal agenesis the surgical intervention was the only option for creating the neovagina.
Line 246: We did not change or improve the Mc Indoe procedure, but we used a bigger vaginal stent, compared to those reported in literature, creating in this way a neovagina of 10.4 cm.
Best regards,
Dr. Adelaida Avino
Round 2
Reviewer 1 Report
The article provides basic information about one type of surgical solution of the creation of the neovagina. After the article was revised, it could be published.